# Novel Relationship between Mitofusin 2-Mediated Mitochondrial Hyperfusion, Metabolic Remodeling, and Glycolysis in Pulmonary Arterial Endothelial Cells

**DOI:** 10.3390/ijms242417533

**Published:** 2023-12-15

**Authors:** Manivannan Yegambaram, Xutong Sun, Alejandro Garcia Flores, Qing Lu, Jamie Soto, Jaime Richards, Saurabh Aggarwal, Ting Wang, Haiwei Gu, Jeffrey R. Fineman, Stephen M. Black

**Affiliations:** 1Center for Translational Science, Florida International University, 11350 SW Village Parkway, Port St. Lucie, FL 34987-2352, USA; myegamba@fiu.edu (M.Y.); xusun@fiu.edu (X.S.); agarciaf@fiu.edu (A.G.F.); qlu@fiu.edu (Q.L.); jamsoto@fiu.edu (J.S.); jairicha@fiu.edu (J.R.); tinwang@fiu.edu (T.W.); haiweigu@asu.edu (H.G.); 2Department of Environmental Health Sciences, Robert Stempel College of Public Health and Social Work, Florida International University, Miami, FL 33199, USA; 3Department of Cellular Biology & Pharmacology, Howard Wertheim College of Medicine, Florida International University, Miami, FL 33199, USA; saurabhaggarwal@uabmc.edu; 4Department of Pediatrics, University of California San Francisco, San Francisco, CA 94143, USA; jeff.fineman@ucsf.edu; 5Cardiovascular Research Institute, University of California San Francisco, San Francisco, CA 94143, USA

**Keywords:** mitofusin, mitochondrial function, glycolysis, metabolomics, pulmonary hypertension

## Abstract

The disruption of mitochondrial dynamics has been identified in cardiovascular diseases, including pulmonary hypertension (PH), ischemia-reperfusion injury, heart failure, and cardiomyopathy. Mitofusin 2 (Mfn2) is abundantly expressed in heart and pulmonary vasculature cells at the outer mitochondrial membrane to modulate fusion. Previously, we have reported reduced levels of Mfn2 and fragmented mitochondria in pulmonary arterial endothelial cells (PAECs) isolated from a sheep model of PH induced by pulmonary over-circulation and restoring Mfn2 normalized mitochondrial function. In this study, we assessed the effect of increased expression of Mfn2 on mitochondrial metabolism, bioenergetics, reactive oxygen species production, and mitochondrial membrane potential in control PAECs. Using an adenoviral expression system to overexpress Mfn2 in PAECs and utilizing ^13^C labeled substrates, we assessed the levels of TCA cycle metabolites. We identified increased pyruvate and lactate production in cells, revealing a glycolytic phenotype (Warburg phenotype). Mfn2 overexpression decreased the mitochondrial ATP production rate, increased the rate of glycolytic ATP production, and disrupted mitochondrial bioenergetics. The increase in glycolysis was linked to increased hypoxia-inducible factor 1α (HIF-1α) protein levels, elevated mitochondrial reactive oxygen species (mt-ROS), and decreased mitochondrial membrane potential. Our data suggest that disrupting the mitochondrial fusion/fission balance to favor hyperfusion leads to a metabolic shift that promotes aerobic glycolysis. Thus, therapies designed to increase mitochondrial fusion should be approached with caution.

## 1. Introduction

Emerging evidence suggests that a contributing factor for pulmonary hypertension (PH) development is the suppression of mitochondria-based respiration and glucose oxidation in favor of aerobic glycolysis (Warburg effect) [1,2]. Previously, we and others have reported that aberrant mitochondrial function leads to energy metabolism disorders in pulmonary arterial endothelial cells (PAECs), increased reactive oxygen species (ROS) generation, and oxidative stress [3,4,5]. Mitochondria constantly undergo fission and fusion, forming a dynamic network to maintain their abundance, shape, and function [6,7]. An imbalance between mitochondrial fusion and fission events is reported for the development of several human diseases, including PH [8,9,10,11,12,13,14,15]. Endothelial cell (EC) dysfunction is one of the first triggers initiating vascular remodeling in PH, and is often accompanied by mitochondrial impairment and dysfunction [16]. Early EC apoptosis in response to injury is a transient event in PH. However, EC injury triggers the activation of several cellular signaling pathways in the endothelium, resulting in uncontrolled proliferation and contributing to vascular remodeling in PH [17]. Mitochondrial dysfunction can result from the dysregulation of mitochondrial fission/fusion dynamics [18,19]. Induction of specific cellular stress can also disrupt the fission/fusion balance and drive mitochondrial hyperfusion (formation of interconnected networks) [20,21,22]. Mitochondrial hyperfusion occurs when the outer and inner membranes of two distinct mitochondria merge, and this process is regulated by GTPases, including mitofusin1 (Mfn1), mitofusin2 (Mfn2), and optic atrophy1 (Opa1) [23,24,25]. Mitofusins are critical for maintaining the mitochondrial network and cellular bioenergetics and are critical regulators required for cell proliferation, apoptosis, and differentiation in many cell types [26,27,28,29,30,31,32]. The outer mitochondrial membrane GTPases mitofusin 1 (Mfn1) and mitofusin 2 (Mfn2) in conjunction with optic atrophy 1 (Opa1), are responsible for the mitochondrial fusion process [23,24,25]. Mfn2 is especially vital because it regulates mitochondria fusion, ER–mitochondria contacts, cell metabolism, apoptosis, and autophagy [33]. 

Mitochondrial dysfunction is typically characterized by an inadequate number of mitochondria, unavailability of necessary substrates to mitochondria, and/or loss of electron transport chain (ETC) efficiency leading to reduced production of adenosine-5′-triphosphate (ATP) [34]. To maintain mitochondrial homeostasis, cells produce new mitochondria through fission and remove dysfunctional mitochondria through mitophagy. In active cells, most ATP synthesized during glucose metabolism is produced in the mitochondria through oxidative phosphorylation. Glucose metabolism initiates in the cell’s cytoplasm, where glucose is first metabolized to pyruvate. Pyruvate is then shuttled to the mitochondria, where it is converted to acetyl-CoA (and NADH from NAD+) by the pyruvate dehydrogenase (PDH) complex [35]. Acetyl-CoA then enters the tricarboxylic acid cycle (TCA cycle). NADH, therefore, serves as a central hydride donor to ATP synthesis through mitochondrial oxidative phosphorylation [36]. Therefore, the ability to produce ATP directly depends on functional mitochondria’s ability to convert the energy of metabolites to reduced NADH and transfer electrons from NADH to the electron transport chain. In PH, glucose metabolism predominantly deviates from mitochondrial oxidative phosphorylation to cytoplasmic glycolysis, generating high levels of pyruvate and lactate [37,38,39,40]. Previously, nontargeted metabolomics and proteomics studies have reported alterations in the TCA cycle pathway in PAECs isolated from PH patients [41]. Further, other metabolomics studies have reported higher levels of TCA cycle metabolites, including citrate, isocitrate, cis-aconitate, succinate, and malate in PH plasma and lung tissue [40,42]. However, the connection between some of the higher levels of TCA cycle metabolites and their contribution to the development of PH is poorly understood.

Mitochondrial dysfunction/damage significantly contributes to many cardiovascular diseases, including PH [38,43,44,45,46,47,48]. Previously, we and others have reported that Mfn2 levels are decreased, which correlates with fragmented mitochondria in both animal models and humans with PH [49,50,51]. Further, we also showed that restoring Mfn2 in PH PAECs increased mitochondrial fusion, enhanced the mitochondrial membrane potential, decreased mitochondrial ROS levels, and enhanced mitochondrial bioenergetics [49]. Adenovirus-mediated Mfn2 overexpression reduces hemodynamic severity and improves exercise capacity in animal models of PH [50]. However, the potential effects of increased Mfn2 expression on metabolic function in normal PAECs is unresolved. Thus, in this study, we overexpressed Mfn2 in PAECs from control lambs and determined its effects on cellular metabolism. Interestingly, our data suggest that the overexpression of Mfn2 induces metabolic reprogramming in PAECs that results in a Warburg phenotype. The increase in aerobic glycolysis was associated with increased levels of hypoxia-inducible factor 1α (HIF-1α) protein and the generation of higher levels of mitochondrial (mt)-ROS. Thus, although prior work has supported that targeting mitochondrial dynamics by regulating Mfn2 might be a potential therapeutic strategy for PH, results from our study indicate that increases in mitochondrial fusion can also produce a glycolytic phenotype, induce metabolic reprogramming, and elevate mitochondrial ROS similar to as observed in PH. Thus, using therapies to increase mitochondrial fusion should be approached cautiously.

## 2. Results

### 2.1. Overexpression of Mfn2 Induces Metabolic Reprogramming in Pulmonary Arterial Endothelial Cells

To understand the effect of increased mitochondrial fusion on cellular metabolism, we overexpressed Mfn2 in PAECs using an adenoviral vector (Ad-MFN2). Initial Western blot studies indicated that multiplicity of infection (MOI) = 5 significantly increased Mfn2 protein in PAECs (Figure 1A). Following detection of the Mfn2 protein, the membrane was stripped and reprobed with an Mfn1 antibody to analyze the expression of Mfn1 protein. Overexpression of Mfn2 increased the Mfn1 protein level in PAECs (Figure 1A). Next, we checked if Mfn2 overexpression affected other fusion/fission mediators. Western blot analysis indicated no change in expression levels of the fusion mediator (Opa1) or the fission mediators Drp1 and Fis1 (Figure 1B–D). Next, utilizing fluorescent microscopy, we observed that Mfn2 overexpression increased the mitochondrial aspect ratio in PAECs, indicative of increased mitochondrial fusion (Figure 1E). In another experiment, TUNEL staining of Mfn2-overexpressing PAECs showed increased apoptosis when compared to control PAECs (Figure 1F). Further, an assessment of cell proliferation utilizing BrdU cell proliferation assay revealed that MFN2 overexpression reduced the proliferation of PAECs (Figure 1G).

We next investigated the consequence of increases in Mfn2 on key cellular metabolites in PAECs using ^13^C-metabolic flux analyses to probe cellular metabolism changes and elucidate metabolic fluxes in the TCA cycle and the linked glutaminolysis pathway. Flux and tracing results were analyzed using the Metaboanalyst platform [52]. Using GC-MS, we validated the measurements of 28 intracellular metabolites (linked to the TCA and glutaminolysis pathway). Both principal component analysis (PCA) (Figure 2A) and partial least-squares discriminant analysis (PLS-DA) plots (Figure 2B) showed a clear cluster distinction between control and Mfn2-overexpressed groups. Values from mass spectrometry analysis were used to develop a heatmap depicting the 28 metabolites altered by Mfn2 overexpression (Figure 2C,D). Out of the 28 metabolites, 12 metabolites were significantly higher, and 3 metabolites were significantly lower in Mfn2-overexpressed PAECs. Importantly, Mfn2 overexpression in PAECs significantly impacted pyruvate and lactate metabolism (Figure 2E,F). Among the TCA and TCA cycle-related metabolites, citrate, malate, and aspartate were significantly increased (Figure 2G–I). Conversely, the α-ketoglutarate, glutamate, and proline levels were significantly lower in Mfn2-overexpressed PAECs (Figure 2J–L). Levels of succinate and oxaloacetate were unchanged in Mfn2-overexpressed PAECs (Figure 2M,N). 

Based on the ^13^C-glucose flux profile with decreased glutamate and α-ketoglutarate levels upon Mfn2 overexpression, we investigated if the PAECs were dependent on glutaminolysis with or without the involvement of the reductive TCA cycle metabolism (i.e., the oxidative TCA cycle running in reverse for glutamine and glutamate metabolism). To accomplish this, PAECs were cultured with media containing ^13^C_5_-glutamine and GC-MS analyzed the extracted TCA cycle intermediates. Results from the ^13^C-glutamine flux analysis indicated no change in TCA cycle intermediates (Figure 3A) and glutamate levels (Figure 3B) upon Mfn2 overexpression. Within the reductive carboxylation pathway, levels of α-ketoglutarate, isocitrate and citrate remained unchanged (Figure 3C–E). Similarly, the level of TCA cycle intermediates involved in glutamine oxidation, including succinate, malate, and oxaloacetate, remained unchanged (Figure F–H). The level of aspartate did not change (Figure 3I). Next, we checked the expression levels of enzymes that are active in reductive carboxylation, including glutamate dehydrogenase (GLUD), aconitase 2 (ACO2), and isocitrate dehydrogenase 2 (IDH2). Western blot analysis revealed that Mfn2 overexpression did not alter GLUD, ACO2, or IDH2 enzyme levels (Figure 3J,L). Further, the oxoglutarate dehydrogenase (OGDH) expression level active in glutamine oxidation remained unchanged (Figure 3M). Our ^13^C-metabolic flux analyses indicated changes in glucose oxidation but not glutamine oxidation/reductive carboxylation in Mfn2-overexpressed cells.

### 2.2. Mfn2 Overexpression Reprograms Cellular Energy Metabolism to Favor Glycolysis in Pulmonary Arterial Endothelial Cells

Our metabolomics data generated a metabolic profile overview that suggested Mfn2 overexpression mediates a shift towards a Warburg phenotype with increased lactate production. To follow up on this, we analyzed the effect of Mfn2 overexpression on cellular ATP production rate and its cellular source. In Mfn2-overexpressed PAECs we identified a significantly increased total ATP production rate (Figure 4A). This was associated with a reduction in mitochondrial ATP production (Figure 4B) and an increase in glycolytic ATP production rate (Figure 4C). Overall, these changes significantly decreased the ATP rate index, i.e., the ratio of mitochondrial ATP production to glycolytic ATP production (Figure 4D) indicative of a Warburg phenotype, which is consistent with the findings in metabolic flux data.

Seahorse extracellular acidification rate (ECAR) analysis (Figure 5A) from the Glycolysis Stress Test revealed increased basal glycolysis (Figure 5B), higher reserve glycolytic capacity (Figure 5C), and elevated maximum glycolytic capacity (Figure 5D). Conversely, oxygen consumption rate (OCR) analysis (Figure 6A) from the Cell Mito Stress Test showed that basal respiration was increased in Mfn2-overexpressed PAECs (Figure 6B), and the amount of oxygen consumed for ATP generation was also increased (Figure 6C). At the same time, the reserve capacity was reduced (Figure 6D), while the maximal respiratory capacity was marginally increased (Figure 6E). Mfn2 overexpression also increased the proton leak, another sign of mitochondrial damage (Figure 6F). These data confirm our ATP rate data and support the conclusion that Mfn2 overexpression induces a Warburg phenotype in control PAECs. Conversely, the oxygen consumption rate (OCR) analysis (Figure 6A) from the Cell Mito Stress Test showed that basal respiration was increased in Mfn2-overexpressed PAECs (Figure 6B), and the amount of oxygen consumed for ATP generation was also increased (Figure 6C). At the same time, the reserve (Figure 6D) and maximal (Figure 6E) respiratory capacities were reduced. Mfn2 overexpression also increased proton leak, another sign of mitochondrial damage (Figure 6F). These data confirm our ATP rate data and support the conclusion that Mfn2 overexpression induces a Warburg phenotype in control PAECs.

### 2.3. Mfn2 Overexpression Increases Mitochondrial (mt)-ROS and HIF-1α Levels in Pulmonary Arterial Endothelial Cells

Western blot analysis revealed that the overexpression of Mfn2 in PAECs increased hypoxia-inducible factor-1α (HIF-1α) levels (Figure 7A). Previously, we have shown that increased mt-ROS activates HIF-1α [53]. In this study, we found that Mfn2 overexpression significantly increased mt-ROS levels in PAECs (Figure 7B). Further, Mfn2 overexpression also affected the mitochondrial membrane potential (Figure 7C). Together, these data link the increase in mt-ROS production with HIF-1α-mediated induction of the Warburg phenotype of PAECs.

## 3. Discussion

Understanding the cellular and molecular mechanisms involved in the pathogenesis of PH is critical for developing drugs to treat the disease. Pulmonary vascular cells harbor abundant mitochondria required for cellular respiration. Further, these mitochondria also modulate many cellular functions, including proliferation, apoptosis, generation of ROS, and intracellular calcium homeostasis. Damaged/dysfunctional mitochondria lead to loss of efficiency in the ETC, alterations in levels of critical metabolites, and reductions in the synthesis of high-energy molecules e.g., ATP. Compelling evidence indicates that mitochondrial dysfunction contributes to PH. PH development is thought to be initiated by an injury to the pulmonary vasculature, resulting in a dysfunctional endothelium [54,55]. The initial stimulus and molecular mechanisms in ECs in PH are yet to be precisely elucidated. However, injury to the endothelium is believed to initiate apoptosis and destabilize the pulmonary vascular intima [54]. Later, EC dysfunction leads to the activation of several cellular signaling pathways in the endothelium, resulting in uncontrolled proliferation, eventually contributing to vascular remodeling and the occlusion of the pulmonary blood vessels [17]. Previously, it has been demonstrated that vascular cells isolated from PH patients have fragmented mitochondria [15,37,56]. This is associated with decreased Mfn2 protein levels and mRNA expression as well as increased expression of the fission proteins dynamin-related protein1 (Drp1) and mitochondrial fission 1 protein (Fis1) [57]. However, Opa1 and Mfn1 levels remained unchanged. Interestingly, in the Su5416/hypoxia-induced PH rat model, both Opa1 and Mfn2 were significantly down-regulated in right ventricular tissues [58]. This suggests that although a delicate mitochondrial fusion/fission balance is maintained by the fusion and fission mediators crucial for cell survival and optimal functioning, they cannot compensate for the loss of individual mediators in PH as seen in different tissue types and organisms. Since decreased levels of Mfn2 contribute to mitochondrial fragmentation in PH, an adenovirus-mediated Mfn2 overexpression was previously evaluated. Mfn2 overexpression inhibited pulmonary artery smooth muscle cells (PASMC) proliferation, enhanced PASMC apoptosis, increased mitochondrial fusion, significantly regressed PH in rodent animal models [50], and improved mitochondrial function in PAECs isolated from PH animals [49]. However, Mfn2 overexpression could disturb the mitochondrial fusion/fission balance. Therefore, we aimed to evaluate the potential negative effects of overexpressing Mfn2 on fusion and fission mediators, mitochondrial metabolism, and bioenergetics in normal PAECs. Three crucial findings arise from our studies. First, ^13^C metabolic flux analysis revealed that the overexpression of Mfn2 in control PAECs significantly impacted cellular metabolism. Second, the Mfn2-mediated mitochondrial hyperfusion significantly attenuated mitochondrial bioenergetics and increased aerobic glycolysis. Third, the Warburg effect in Mfn2-overexpressed cells was associated with increased mt-ROS generation. Until now, it has been thought that only mitochondrial hyperfission is linked with mitochondrial dysfunction, mt-ROS generation, and glycolytic metabolic reprogramming. For the first time, this study confirms that mitochondrial hyperfusion, caused by Mfn2 overexpression, leads to a similar functional outcome, signifying the pivotal role of mitochondrial network dynamics homeostasis on endothelial metabolism.

Mfn2 deficiency leads to increased mitochondrial fission and hyperproliferation of PASMC in human PH and rodent PH models [50]. In this study, overexpression of Mfn2 in control PAECs increased mitochondrial fusion, increased apoptosis, and decreased cell proliferation. This is supported by a previous study showing MFN2 overexpression increased the expression levels of autophagy-associated proteins, including Beclin1, LC3B II/soluble p62, mito-p62, mito-PINK1, and mito-Parkin [59]. Conversely, Mfn2 deficiency has been shown to reduce autophagy and impair mitochondrial quality [60]. Interestingly, the knockdown of MFN2 leads to a decrease in the generation of ROS and reduced expression of components of the respiratory chain and transcription factors associated with oxidative metabolism [61]. Taken together, these data suggest that mitochondrial dynamics, particularly those mediated by Mfn2, play an important role in endothelial cell function and viability. Many studies have identified shifts in cellular metabolism in PH PAECs and PASMC, mainly demonstrating less oxidative metabolism of glucose [38,62,63]. Previously, our studies have also reported a distinct elevated lactate/pyruvate ratio in lung tissues from ovine animal models of PH [64,65]. For the first time, our metabolomic flux analysis using ^13^C-glucose tracers reveals a change in mitochondrial metabolism with the overexpression of Mfn2 in control PAECs and, importantly, links mitochondrial hyperfusion to glycolysis. Analysis of the TCA and TCA cycle-related metabolites suggested a stimulation of pyruvate/lactate metabolism with concurrent increased TCA intermediates. Previous metabolomics studies have reported a similar disruption of the TCA cycle pathway with higher metabolites, including citrate, isocitrate, cis-aconitate, succinate, and malate in PH plasma and lung [40,42]. An increase in TCA cycle intermediates and a decrease in levels of glutamate and α-ketoglutarate indicated the involvement of glutamine metabolism (glutaminolysis) in maintaining TCA cycle function. During glutaminolysis, cells utilize glutamate (generated from glutamine), which is converted to α-ketoglutarate by glutamate dehydrogenase and metabolized in the TCA cycle in the oxidative (clockwise) or reductive (anti-clockwise) direction to generate a citrate [66]. However, metabolic flux analyses using ^13^C-glutamine tracers indicated that MFN2 overexpression did not alter glutamine oxidation or reductive carboxylation. This suggests that MFN2 exerts its effect only on glucose oxidation and that this may be sufficient to maintain the TCA cycle function and cell survival during Mfn2-mediated hyperfusion. Results from our ^13^C metabolic flux data are supported by another study, which reported that cardiac-specific MFN2-deficient mice have significantly reduced levels of intermediates of the TCA cycle and fatty acid β-oxidation [67]. Importantly, our ^13^C metabolic flux data demonstrate that stimulating mitochondrial hyperfusion can also lead to glycolytic metabolic alterations, which has only previously been reported with mitochondrial fission in PH [57,68]. However, a comprehensive assessment of different substrate uptake and how the PAECs further utilize these substrates during mitochondrial fission and fusion events must be carried out to characterize substrate utilization in PH fully. Our ATP rate assays also revealed that Mfn2 overexpression decreased the mitochondrial ATP production rate. However, the total ATP rate was increased, indicating a shift from oxidative phosphorylation to glycolysis. Thus, unlike pulmonary vascular cells that rely primarily on mitochondrial OXPHOS for energy production, the Mfn2-overexpressing PAECs behaved like cancer cells and used glycolysis. This was confirmed by analyzing the extracellular acidification rate, where increasing MFN2 expression enhanced cellular glycolysis as evidenced by basal, reserve, and maximal glycolytic capacity increases. Interestingly, previously it was reported that the targeted deletion of Mfn2 in myoblasts significantly increases maximal respiration and spare respiratory capacity [69], suggesting that Mfn2 levels modulate mitochondrial bioenergetics. Further, these data suggest that any imbalance in mitochondrial dynamics disturbs the mitochondrial bioenergetics. Thus, therapies targeted at mitochondrial fission or fusion must be carefully titrated to match normal physiologic levels and restore normal cellular metabolism.

In this study, we were able to link Mfn2-mediated mitochondrial hyperfusion to increased HIF-1α accumulation. HIF-1α is a central transcriptional regulator of cellular metabolism [70,71]. HIF-1α induces the expression of all glycolytic enzymes, including HK-2 (83–85) and glucose transporters, driving the glycolytic cascade that converts glucose into pyruvate. Pyruvate can then either be converted into acetyl-CoA for further metabolism in the TCA cycle or be converted into lactate. Under hypoxic conditions, HIF-1α favors lactate production while inhibiting pyruvate entry into the TCA cycle. Previously, it has been reported that ECs isolated from patients with idiopathic pulmonary arterial hypertension (IPAH-ECs) have increased HIF-1α expression [72]. Another study reported that IPAH-ECs show an abnormal metabolic phenotype characterized by low numbers of mitochondria, significantly higher glycolytic rate, and substantial changes in bioenergetics [38]. However, it is unclear if the low numbers of mitochondria observed in IPAH-ECs were possibly due to increased mitofusins 1/2 mediated mitochondrial fusion and/or decreased fusion mediators. However, further studies will be required to investigate the link between abnormal HIF-dependent signaling and fission/fusion machinery imbalance. Multiple factors, including mitochondrial enzymes and the respiratory complex, can influence the activity of HIF-1α [73]. We, and others, have shown that mt-ROS levels can regulate HIF-1α levels [74]. An increase in oxidative stress is observed in the lungs and pulmonary vasculature of animals and humans with PH associated with increased ROS generation from several sources, including the mitochondria [75,76,77,78,79]. Mitochondrial fragmentation has been linked to an increase in ROS production [80], and there is also evidence of the contribution of Mfn2 to ROS production and inflammation [81]. Indeed, in our study, we identified increased levels of mt-ROS in Mfn2-overexpressing PAECs that correlated with increased HIF-1α levels. Further, our recent work has shown that normalizing mitochondrial network dynamics decreases mt-ROS-mediated increases in HIF-1α and attenuates cellular glycolysis [49]. We have previously shown that the PH stimulus, endothelin-1, increases HIF-1 signaling in PAECs via mt-ROS and also enhances the HIF-1-dependent downstream target genes, enolase-2 and glucose-6-phosphate dehydrogenase, which are known to be involved in glycolysis [53]. Further, treatment with mitochondrial-targeted antioxidant (MitoQ), not only decreased endothelin-1-induced mt-ROS in PAECs, but also attenuated the ROS-mediated increase in HIF-1-dependent promoter activity [53]. Thus, it is likely that mt-ROS-mediated activation of HIF-1α is the mechanism by which MFN2-mediated hyperfusion stimulates aerobic glycolysis. Together, these data suggest that modulating mt-ROS levels could be a therapeutic target for PH alone or in combination with a regulator of mitochondrial dynamics such as mdivi-1, a Drp1 inhibitor [80], or MiM111, a fusion burst activator [82]. Indeed, the therapeutic potential of Mfn2 overexpression to suppress PH development in rodent models was reported [50]. However, contradictory studies indicate that Mfn2 can induce PASMC proliferation in hypoxic PH [32]. Further, MFN2 knockout mice exhibit significantly better cardiac function than control mice [67]. Thus, further studies will be required to understand the molecular mechanisms and setpoints involved in regulating mitofusin-mediated metabolic reprogramming and how disrupting these leads to enhanced glycolytic flux.

## 4. Materials and Methods

### 4.1. Reagents

All reagents, unless specified, were obtained from Sigma-Aldrich (St. Louis, MO, USA). Reagents and supplies for SDS-PAGE electrophoresis and Western blots were purchased from Thermo Fisher Scientific (Waltham, MA, USA) and BioRad (Hercules, CA, USA). For Western blots, a primary monoclonal Mfn2 antibody (#11925S, Cell Signaling Technologies, Danvers, MA, USA), a primary polyclonal Mfn1 antibody (#PA5-79665, Thermo Fisher Scientific, Waltham, MA, USA), a primary monoclonal Opa1 antibody (#612606, BD Biosciences, Franklin Lakes, NJ, USA), a primary monoclonal Drp1 antibody (#5391S, Cell Signaling Technologies, Danvers, MA, USA), a primary monoclonal Fis1 antibody (#32525S, Cell Signaling Technologies, Danvers, MA, USA), a primary monoclonal HIF-1α antibody (#NB100-105SS, Novus Biologicals, Centennial, CO, USA), a primary monoclonal GLUD1/2 antibody (#12793S, Cell Signaling Technologies, Danvers, MA, USA), a primary monoclonal IDH2 antibody (#56439S, Cell Signaling Technologies, Danvers, MA, USA), a primary monoclonal ACO2 antibody (#6571S, Cell Signaling Technologies, Danvers, MA, USA), a primary monoclonal OGDH antibody (#26865S, Cell Signaling Technologies, Danvers, MA, USA), and a primary monoclonal β-Actin antibody (#A3854-200UL, Sigma, St. Louis, MO, USA), were used. The Western blot secondary antibodies used were an anti-mouse IgG, HRP-linked antibody (#7076P2), and an anti-rabbit IgG, HRP-linked antibody (#7074P2). These were purchased from Cell Signaling Technologies (Danvers, MA, USA). All the Seahorse reagents were purchased from Agilent Technology (Santa Clara, CA, USA).

### 4.2. Animal and Ethics Statement

The Committee on Animal Research at the Florida International University and the University of California, San Francisco, approved the animal protocols and procedures used to procure the pulmonary arterial endothelial cells (PAECs) from control lambs. All experiments were performed according to the university guidelines that comply with national and international regulations, as previously described [83].

### 4.3. Cell Culture and Adenoviral Transduction of PAECs

PAECs were isolated and cultured in the laboratory following our previously published protocol [83,84]. Adenoviruses (Ad) Ad-MFN2 and Ad-GFP (control) were purchased from Vector Biolabs (Malvern, PA, USA). PAECs were grown with Dulbecco’s Modified Eagle Medium (DMEM) with 2% fetal bovine serum (FBS) (Corning, NY, USA) in 10 cm culture plates. Upon reaching 70% confluency, cells were counted using a hemocytometer. After cell count measurement, PAECs were replated and allowed to attach on 6-well plates. PAECs were transduced with Ad-MFN2 (MOI = 5) or Ad-GFP (control, MOI = 5) and were allowed to grow for 8 h at 37 °C in an incubator with 5% CO_2_. Post transduction, the media were replaced with complete media, and the cells were allowed to grow for 40 h at 37 °C in an incubator with 5% CO_2_.

### 4.4. Western Blot Analysis

PAECs were grown in 6-well culture plates and transduced with either Ad-MFN2 or Ad-GFP (control) as described above. The cells were lysed using RIPA Lysis and Extraction Buffer (Thermo Scientific, Waltham, MA, USA), and the protein concentration was determined by BCA assay (Thermo Scientific, Waltham, MA, USA). Protein extracts (20 μg) were separated on either 7.5% or 4–20% polyacrylamide gels (Bio-Rad, Hercules, CA, USA) at 150 V for 60 min using a PowerPac Basic (Bio-Rad, Hercules, CA, USA). All gels were electrophoretically transferred to Trans-Blot Turbo Mini 0.2 µm PVDF membranes (Bio-Rad, Hercules, CA, USA). The membranes were blocked with 5% nonfat dry milk in Tris-buffered saline (TBS) (Bio-Rad, Hercules, CA, USA) containing 0.1% Tween for 1 h. After being blocked, the membranes were washed three times in TBS containing 0.1% Tween for 5 min at room temperature and incubated with the recommended dilution of a specific monoclonal antibody for 16 h at 4 °C. After being washed three times in TBS containing 0.1% Tween for 5 min at room temperature, the membranes were incubated with an appropriate secondary antibody conjugated with horseradish peroxidase (1:1000) for 1 h at room temperature. Membranes were washed three times in TBS containing 0.1% Tween for 5 min at room temperature. The protein bands were visualized by chemiluminescent procedures using SuperSignal West Femto Maximum Sensitivity Substrate (Thermo Scientific, Waltham, MA, USA), and band intensities were determined using the Odyssey XF imaging system (Lincoln, NE, USA).

### 4.5. Gas Chromatography–Mass Spectrometry (GC-MS) Metabolic Flux Assay

Metabolic flux assay followed our previously published protocols [85,86]. PAECs were grown in 6-well culture plates and transduced with either Ad-MFN2 or Ad-GFP (control) and were allowed to grow for 32 h at 37 °C in an incubator with 5% CO_2_. For the ^13^C_6_-glucose flux experiments, the culture medium was replaced with 5 mM U-^13^C_6_-glucose (Cambridge Isotope Laboratories, Tewksbury, MA, USA) in DMEM without glucose, glutamine, and pyruvate (Thermo Scientific, Waltham, MA, USA) supplemented with 10% dialyzed FBS (Thermo Scientific, Waltham, MA, USA), 1% penicillin-streptomycin (Thermo Scientific, Waltham, MA, USA), and 1% L-glutamine (Thermo Scientific, Waltham, MA, USA) and allowed to grow for 16 h at 37 °C in an incubator with 5% CO_2_. For the ^13^C_5_-glutamine flux experiments, the culture medium was replaced with 5 mM ^13^C_5_-glutamine (Cambridge Isotope Laboratories, Tewksbury, MA, USA) in DMEM without glucose, glutamine, and pyruvate (Thermo Scientific, Waltham, MA, USA) supplemented with 10% dialyzed FBS (Thermo Scientific, Waltham, MA, USA), 1% penicillin-streptomycin (Thermo Scientific, Waltham, MA, USA), and 1% glucose (Thermo Scientific, Waltham, MA, USA) and allowed to grow for 16 h at 37 °C in an incubator with 5% CO_2_. The next day, cellular extracts from 1 million cells were derivatized following our previously published protocol and injected into the Agilent 7820 GC-5977 MSD system for GC-MS spectral acquisition (Agilent Technologies, Inc., Santa Clara, CA, USA) [85,87]. The GC-MS was run using splitless mode with helium as the carrier gas. One μL of derivatized sample was injected into the system and the chromatographic separation obtained via a Zorbax DB5-MS + 10 m Duragard Capillary Column (30 m × 250 μm × 0.25 μm) (Agilent, Santa Clara, CA, USA). The temperature of the column was held at 60 °C for 1 min before increasing at a rate of 10 °C/min until 325 °C was reached. The temperature of the column was kept at 325 °C for 10 min. Mass spectral signals were recorded in the full scan mode utilizing electron ionization (EI, 70 eV), with a mass range 50–600 Da. Agilent MassHunter software (Version B.09.00) was applied to the raw GC-MS data in a targeted way to monitor compounds. The mass isotopomer distributions (MIDs) for each sample were calculated by integrating metabolite ion fragments, and the IsoCor 2.2.2 software was employed to correct the natural abundance of isotopes.

### 4.6. Metabolic Flux Analysis

Metabolic flux data were analyzed with the MetaboAnalyst platform, as previously described [88]. Data normalization was performed by log transformation and Pareto data scaling. Principal component analysis (PCA) and partial-least-squares discrimination analysis (PLS-DA) were performed for multivariate statistical analysis. Correlation analysis was used to visualize the overall correlations between two groups (control and Mfn2-overexpressed sets), and the clustering result was shown as a heatmap. A summary of enriched metabolites was displayed using the TCA cycle.

### 4.7. Measurement of Oxygen Consumption Rate

The XFe24 Analyzer (Agilent Technologies, Santa Clara, CA, USA) and XF Cell Mito Stress Test Kit (103015-100; Agilent Technologies, Santa Clara, CA, USA) were used for the mitochondrial bioenergetic analyses. PAECs were transduced with either Ad-MFN2 or Ad-GFP (control) in 6-well culture plates and allowed to grow for 24 h at 37 °C in an incubator with 5% CO_2_. The following day, cells were replated onto the XFe24 culture microplates (at the predetermined optimum number of 75,000/0.32 cm^2^ PAECs/well) and allowed to grow for 24 h at 37 °C in an incubator with 5% CO_2_. Cell growth medium for PAECs was exchanged for XF DMEM medium (102353-100; Agilent Technologies, Santa Clara, CA, USA) with 5.5 mM glucose (103577-100; Agilent Technologies, Santa Clara, CA, USA), 2 mM pyruvate (103578-100; Agilent Technologies, Santa Clara, CA, USA), and 2 mM glutamine (103579-100; Agilent Technologies, Santa Clara, CA, USA), pH 7.4. The cell plate was placed in a CO_2_-free incubator at 37 °C for 45–60 min for temperature and pH calibration. For the Mito Stress Test, we sequentially injected Oligomycin (1 μm final concentration), carbonylcyanide 4-(trifluoromethoxy) phenylhydrazone (FCCP, 1 μm final concentration), and Rotenone + antimycin A (0.5 μm final concentration of each) and measured the oxygen consumption rate (OCR). Using these agents, we calculated basal mitochondrial respiration, reserve respiratory capacity, and maximal respiratory capacity measurements in pmols/min of oxygen consumed using the Seahorse XF Cell Mito Stress Test Report Generator from the Wave software 2.6. The Seahorse XFe data were normalized to protein amount. Basal respiration was determined by subtracting measurement 10 (minimum rate after rotenone/antimycin A injection) from measurement 3 (last rate before oligomycin injection). OCR associated with ATP production was determined by subtracting measurement 4 (minimum rate after Oligomycin injection) from measurement 3 (last rate before oligomycin injection). Spare respiratory capacity was determined by subtracting basal respiration from maximal respiration. For control cells, maximal respiration was determined by subtracting measurement 10 (minimum rate after rotenone/ antimycin A injection) from measurement 7 (maximum rate after FCCP injection). For Ad Mfn2-overexpressing cells, maximal respiration was determined by subtracting measurement 10 (minimum rate after rotenone/antimycin A injection) from measurement 9 (maximum rate after FCCP injection). Proton leak was determined by subtracting measurement 10 (minimum rate after rotenone/antimycin A injection) from measurement 4 (minimum rate after Oligomycin injection).

### 4.8. Measurement of Extracellular Acidification Rate

The XFe24 Analyzer (Agilent Technologies, Santa Clara, CA, USA) and XF Cell Glycolysis Stress Test Kit (103020-100; Agilent Technologies, Santa Clara, CA, USA) were used for glycolysis analyses. The measurement of extracellular acidification was obtained from the generation of protons during the conversion of glucose to lactate in glycolysis. PAECs were transduced with either adenoviruses Ad-MFN2 or Ad-GFP (control) in 6-well culture plates and allowed to grow for 24 h at 37 °C in an incubator with 5% CO_2_. The following day, cells were replated onto the XFe24 culture microplates (at the predetermined optimum number of 75,000/0.32 cm^2^ PAECs/well) and allowed to grow for 24 h at 37 °C in an incubator with 5% CO_2_. Cell growth medium for PAECs was exchanged for XF DMEM medium (102353-100; Agilent Technologies, Santa Clara, CA, USA) with 2 mM glutamine (103579-100; Agilent Technologies, Santa Clara, CA, USA), pH 7.4. For the Glycolysis Stress Test, we sequentially injected Glucose (10 mM final concentration), oligomycin (1 µM final concentration), and 2-deoxyglucose (2-DG; 50 mM final concentration). Using these agents, we calculated glycolytic capacity and reserve using the Seahorse XF Glycolysis Stress Test Report Generator from the Wave software. The Seahorse XFe data were normalized to protein amount. Basal glycolysis was determined by subtracting measurement 3 (last rate before glucose injection) from measurement 4 (maximum rate before oligomycin injection). Glycolytic reserve was determined by subtracting maximum glycolytic capacity from basal glycolysis. Maximum glycolytic capacity was determined by subtracting measurement 3 (last rate before glucose injection) from measurement 7 (maximum rate after oligomycin injection).

### 4.9. Analysis of Real-Time Cellular ATP Rates

We utilized an XF Real-Time ATP Rate Kit (Agilent Technologies, Santa Clara, CA, USA) and the Seahorse XFe24 analyzer to estimate changes in cellular total ATP production rates, and the ATP generated from mitochondrial oxidative phosphorylation (OXPHOS) or cellular glycolysis according to the manufacturer’s protocol. Basal OCR and ECAR rates were measured first, followed by oligomycin injection (1.5 µM). This decreased OCR by inhibiting mitochondrial ATP synthesis, which provided mitochondrial ATP production rate quantification. Complete inhibition of mitochondrial respiration with rotenone (0.5 mM) plus antimycin A (0.5 µM) in combination with PER data (combination ECAR data with the buffer factor of the assay medium) provided the calculation of the glycolytic ATP production rate. Mitochondrial ATP production rate was calculated with the equation: OCR_ATP_ (pmols/min)*2(pmol O/pmolO_2_)*P/O (pmol ATP/pmol O), and glycolytic ATP production rate was calculated using the glycolytic PER.

### 4.10. In Situ Apoptosis Assay

We used the terminal deoxynucleotidyl transferase (Tdt)-mediated dUTP nick-end labeling (TUNEL) assay to determine the levels of apoptosis in PAECs. In situ detection of cleaved, apoptotic DNA fragments was performed according to the manufacturer’s instructions (Click-iT™ Plus TUNEL Assay, Thermo Scientific, Waltham, MA, USA). PAECs were transduced with either Ad-MFN2 or Ad-GFP (control) in 6-well culture plates and allowed to grow for 48 h at 37 °C in an incubator with 5% CO_2_. Cells were replated on coverslips in 24-well culture plates, washed with PBS, and fixed with 4% paraformaldehyde for 15 min at room temperature. After removing the fixative, the samples were washed 3 times for 5 min with PBS, and a permeabilization reagent (100% Methanol) was added to the coverslip and incubated for 20 min at −20 °C. Coverslips were washed 3 times with PBS and 100 µL of the TdT reaction buffer was added and incubated for 10 min at room temperature. The reaction buffer was replaced with 100 µL of the TdT reaction cocktail and the coverslips were incubated for 60 min at 37 °C. The coverslips were washed twice with 3% BSA in PBS for 2 min each. Next, 100 µL of the Click-iT^®^ reaction cocktail was added to each coverslip and incubated for 30 min at room temperature. The coverslips were washed with 3% BSA in 1X PBS for 5 min. Next, 500 µL of a 3% BSA in PBS solution was added per sample; then, 1 drop of NucBlue was added per coverslip and incubated for 10 min. After washing with PBS 3 times, the coverslips were mounted on glass slides with Prolong glass antifade mountant. The TdT in situ-labeled immunofluorescent images were observed and imaged with a Keyence microscope (Keyence, Wauwatosa, WI, USA), using a 20× objective with 590 nm excitation and 615 nm emission. The images were processed and analyzed with ImageJ. The frequency of labeled cells was calculated by counting cells on each slide in areas with the highest number of TdT-labeled nuclei. Nuclei with any detectable staining above background and a size above 29 µm^2^ were scored as positive. The ratio of nuclei labeled with TdT to the total nuclei counted was expressed as apoptosis percentage.

### 4.11. BrdU Cell Proliferation Assay

BrdU Cell Proliferation Assay was performed following the manufacturer’s protocol (Cell Signaling Technologies, Danvers, MA, USA). PAECs were grown in 6-well culture plates and transduced with either Ad-MFN2 or Ad-GFP (control) and were allowed to grow for 24 h at 37 °C in an incubator with 5% CO_2_. The following day, PAECs (1 × 10^4^ cells/well) were replated on 96-well plates and BrdU solution was added to the cell culture and further incubated for 24 h at 37 °C in an incubator with 5% CO_2_. Media were removed and fixing/denaturing solution was added to the adherent cells followed by addition of detection antibody solution, HRP-conjugated secondary antibody, and TMB substrate with 30 min incubation time between each step. Finally, absorbance was read at 450 nm using the BioTek Cytation microplate reader (Agilent Technologies, Santa Clara, CA, USA) within 30 min of adding the stop solution.

### 4.12. Analysis of Mitochondrial Fusion

PAECs were grown in 6-well culture plates and transduced with either Ad-MFN2 or Ad-GFP (control) and were allowed to grow for 48 h at 37 °C in an incubator with 5% CO_2_. PAECs were replated on a glass bottom dish and labeled with 30 nM MitoTracker (#M7512, Thermo Scientific, Waltham, MA, USA) for 15 min at 37 °C. Hoechst 33342 (#R37605, Thermo Scientific, Waltham, MA, USA) was used for nuclei staining. Fluorescence images were then captured using the KEYENCE BZ-X800 fluorescence microscope with 60× objective, as described previously [49]. Mitochondrial fusion morphology analysis was performed using ImageJ/FiJi (NIH). The mitochondrial long axis and wide axis were measured by particle analysis measurement. The aspect ratio was calculated to represent the mitochondrial fragmentation characteristics [89].

### 4.13. Analysis of Mitochondrial (mt)-ROS and the Mitochondrial Membrane Potential

PAECs were grown in 6-well culture plates and transduced with either Ad-MFN2 or Ad-GFP (control) and were allowed to grow for 48 h at 37 °C in an incubator with 5% CO_2_. MitoSOX™ Red mitochondrial reactive oxygen species (ROS) indicator (Molecular Probes, Gr and Island, NY, USA) a fluorogenic dye for detection of ROS in the mitochondria of live cells was used. Mitochondrial membrane potential was determined using TMRM (tetramethylrhodamine methyl ester perchlorate, Molecular Probes, Eugene, OR, USA). Cells were incubated in media containing either MitoSOX Red (5 μM) or TMRM (50 nM), for 30 min at 37 °C in the dark. An Olympus IX51 microscope (Olympus life science, Center Valley, PA, USA) equipped with a CCD camera (Hamamatsu Photonics, Bridgewater, NJ, USA) was used to acquire fluorescent images. Fluorescence images were then captured using an excitation of 510 nm and an emission at 580 nm for MitoSOX. In another experiment, fluorescence images were captured using an excitation of 548 nm and an emission at 575 nm for TMRM. The average fluorescent intensities were quantified using Image ProPlus version 5.0 imaging software (Media Cybernetics).

### 4.14. Statistical Analysis

Statistical analysis for this study was performed using GraphPad Prism version 4.01. The mean ± SEM was calculated for all samples, and significance was determined by the unpaired t-test. A statistically significant test result *p* < 0.05 was accepted. For metabolic flux data analysis, Mass isotopomer distribution (MID) and mean enrichment (ME) were calculated for each metabolite after correction for natural abundance using IsoCor software 2.2.2 [90]. A false discovery rate cut-off of 0.05 was used to identify differentiating metabolites.

## 5. Conclusions

This study shows that alterations in mitochondrial dynamics and/or redox state can induce metabolic reprogramming, leading to a switch to aerobic glycolysis (the Warburg effect). Further, our data support the conclusion that a delicate balance between fission and fusion is essential for cell survival and normal metabolic function. Our data show that defects in mitochondrial dynamics severely affect energy production, oxidative stress, and metabolic pathways. However, we also conclude that therapeutically targeting mitochondrial network dynamics must be approached cautiously until we better understand the physiologic setpoints the pulmonary vasculature will tolerate.

## Figures and Tables

**Figure 1 ijms-24-17533-f001:**
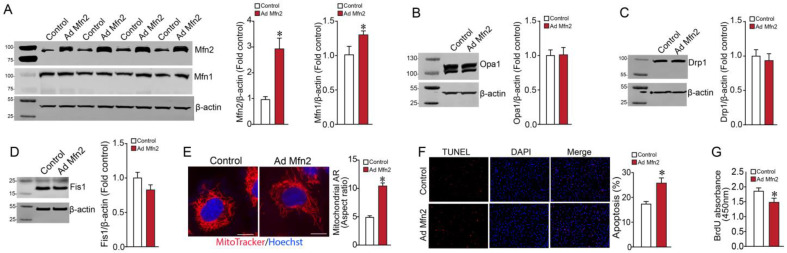
Effect of Mfn2 overexpression on fusion and fission mediators, apoptosis, and proliferation in pulmonary arterial endothelial cells. PAECs were transduced with Ad-MFN2 (MOI = 5) or Ad-GFP, and Western blot analysis was used to confirm increased Mfn2 protein levels ((**A**), upper blot). The membrane was stripped and reprobed with an Mfn1 antibody to analyze the expression of Mfn1 protein. Overexpression of Mfn2 increased Mfn1 protein levels ((**A**), middle blot). Mfn2 overexpression did not affect the expression level of the fusion mediator Opa1 (**B**) or the fission mediators Drp1 (**C**) and Fis1 (**D**). Representative images are shown. β-actin was used to normalize protein loading. Mfn2 overexpression increased the aspect ratio (AR) in PAECs (**E**). TUNEL staining of PAECs identified increased apoptosis in MFN2-overexpressing PAECs (**F**). MFN2 overexpression also reduced the proliferation of PAECs (**G**). Data are mean ± SEM; n = 4–6. * *p* < 0.05 vs. control PAECs.

**Figure 2 ijms-24-17533-f002:**
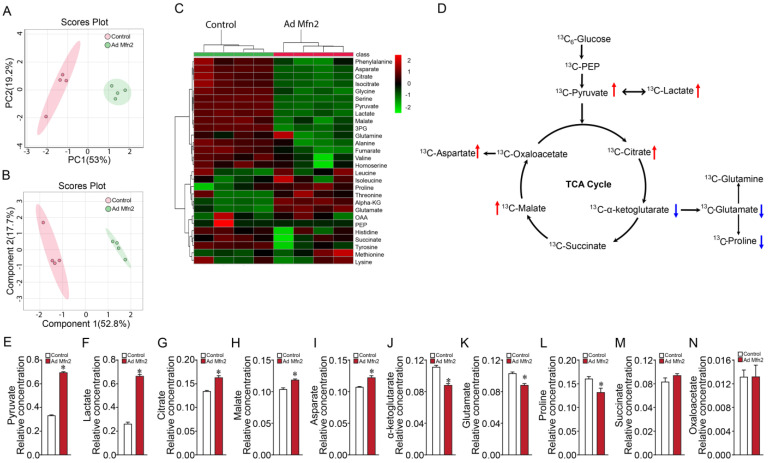
Mfn2 overexpression causes metabolic reprogramming in pulmonary arterial endothelial cells. PCA (**A**) and PLS-DA (**B**) score plots generated using MetaboAnalyst revealed separation in metabolite profiles between control and Mfn2-overexpressing PAECs. A heatmap illustrates the changes in metabolite levels in Mfn2-overexpressing PAECs (**C**). An overview of the TCA cycle and linked glutaminolysis metabolic pathways (**D**). The notations are as follows: the red arrow indicates significantly upregulated metabolite in Mfn2-overexpressed PAECs vs. control PAECs group; the blue arrow indicates downregulated metabolite in Mfn2-overexpressed PAECs vs. control PAECs group. Mfn2 overexpression in PAECs significantly increased pyruvate and lactate production (**E**,**F**). Among the TCA cycle and TCA cycle-related metabolites, citrate, malate, and aspartate levels were increased (**G**–**I**). The levels of α-ketoglutarate, glutamate, and proline were significantly decreased (**J**–**L**) in Mfn2-overexpressed PAECs. Succinate and oxaloacetate levels were unchanged by Mfn2 overexpression (**M**,**N**). Data are mean ± SEM; n = 4. * *p* < 0.05 vs. control PAECs.

**Figure 3 ijms-24-17533-f003:**
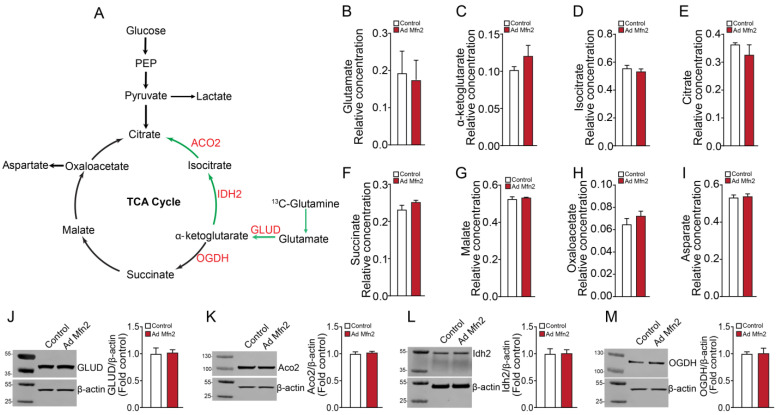
Mfn2 overexpression does not alter glutamine oxidation or reductive carboxylation in pulmonary arterial endothelial cells. An overview of the TCA cycle and linked glutaminolysis metabolic pathway (green arrow indicates reductive carboxylation pathway) (**A**). The levels of TCA cycle and TCA cycle-related metabolites, including glutamate, α-ketoglutarate, isocitrate, citrate, succinate, malate, oxaloacetate, and aspartate, were unchanged upon Mfn2 overexpression (**B**–**I**). Mfn2 overexpression did not affect the levels of GLUD (**J**), ACO2 (**K**), IDH2 (**L**), and OGDH (**M**). Data are mean ± SEM; n = 4.

**Figure 4 ijms-24-17533-f004:**
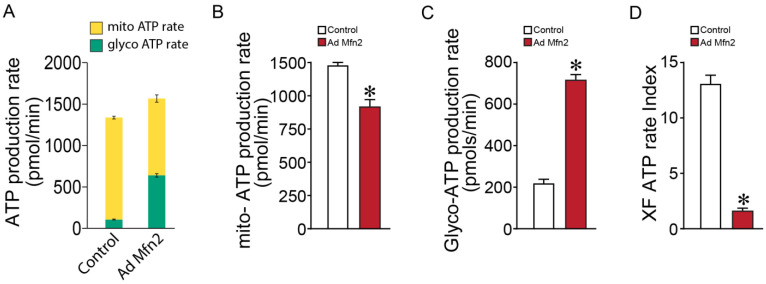
Mfn2 overexpression alters the cellular ATP production rate in pulmonary arterial endothelial cells. Mfn2 overexpression increased the total cellular ATP production rate (**A**). The mitochondrial ATP production rate was reduced (**B**), and the glycolytic ATP production rate was increased (**C**). The ATP rate index was decreased (**D**). Data are mean ± SEM; n = 10. * *p* < 0.05 vs. control PAECs.

**Figure 5 ijms-24-17533-f005:**
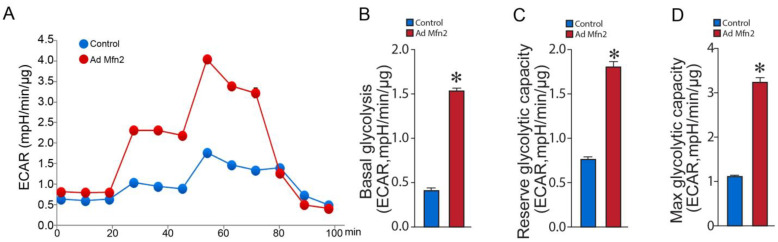
Mfn2 overexpression stimulates cellular glycolysis in pulmonary arterial endothelial cells. Mfn2 overexpression altered the extracellular acidification rate (ECAR) profile in PAECs (**A**). Mfn2 overexpression increased basal glycolysis (**B**), elevated the reserve glycolytic capacity (**C**), and increased the maximum glycolytic capacity (**D**). Data are mean ± SEM; n = 10. * *p* < 0.05 vs. control.

**Figure 6 ijms-24-17533-f006:**
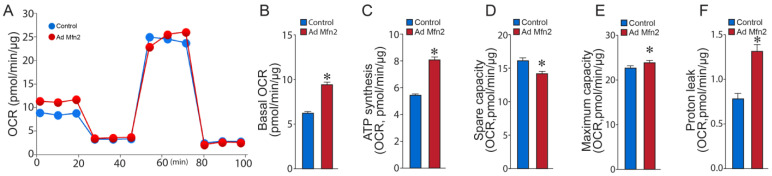
Mfn2 overexpression disrupts mitochondrial bioenergetics in pulmonary arterial endothelial cells. Mfn2 overexpression disrupts the bioenergetic profile for oxygen consumption rate (OCR) (**A**) such that basal respiration was increased (**B**), the OCR for ATP synthesis was increased (**C**). The reserve capacity was decreased (**D**) while the maximum respiratory capacity (**E**) and the proton leak were increased (**F**). Data are mean ± SEM; n = 10. * *p* < 0.05 vs. control PAECs.

**Figure 7 ijms-24-17533-f007:**
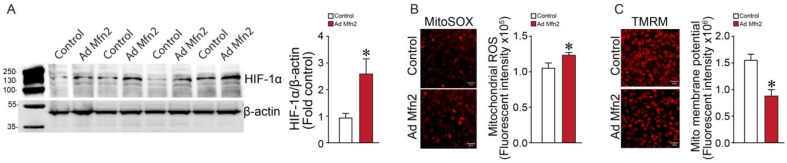
Mfn2 overexpression increases mitochondrial (mt)-ROS and HIF-1α levels in pulmonary arterial endothelial cells. Mfn2 overexpression increased HIF-1α protein levels in PAECs (**A**). A representative image is shown. β-actin was used to normalize protein loading. Mitochondrial-ROS levels are increased (**B**) and the mitochondrial membrane potential is decreased (**C**) by Mfn2 overexpression. Data are mean ± SEM; n = 4–5. * *p* < 0.05 vs. control PAECs.

## Data Availability

Original data are available on request to the corresponding author.

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
