# Peer review of "Novel Relationship between Mitofusin 2-Mediated Mitochondrial Hyperfusion, Metabolic Remodeling, and Glycolysis in Pulmonary Arterial Endothelial Cells"

_ijms, 2023, doi:10.3390/ijms242417533_

Round 1
Reviewer 1 Report
Comments and Suggestions for Authors
In this manuscript, Yegambaram et al., evaluate the overexpression of mitofusin 2 (Mfn2) in pulmonary arterial endothelial cells (PAECs) derived from a pulmonary hypertension model. Although the work seems interesting and the experiments well designed, the conclusions they reach are not supported by the established methods and the results presented. In this sense, the following points are requested to be clarified and explained in the manuscript:
How to distinguish overexpression of Mfn2 from restoring its expression in the evaluated model? since it is mentioned that these PAECs present reduced levels of Mfn2, however throughout the text the "overexpression" and "restoration" of Mfn2 levels are mentioned as if they were the same thing.
The authors mention that "Three crucial findings arise from our studies": 1) "overexpression of Mfn2 in PAECs induces mitochondrial hyperfusion", 2) mitochondrial hyperfusion significantly attenuates mitochondrial bioenergetics and increases aerobic glycolysis". However this mitochondrial hyperfusion is not demonstrated in any of their results. Although the presence of Mfn2 is highly associated with mitochondrial fusion, it also regulates other mechanisms, which are mentioned by the authors (line 54). If mitochondrial hyperfusion is not experimentally demonstrated in this manuscript, it is necessary to change the title and the sentences under discussion that assert this fact. 3) "this Warburg effect was induced by mt-ROS-mediated activation of HIF-1α protein levels". In this regard, no measurement is made to demonstrate that mt-ROS are mediating glycolysis or altering HIF-1α levels, remaining only speculation.
minor comments:
chance mi-ROS by mt-ROS (line 225)
XF24 Analyzer and XFe24 analyzer are the same? (lines 341, 346)
line 367: What result were 3 groups evaluated on? that have been compared with ANOVA?
Author Response
Reviewer 1
Comments and Suggestions for Authors
In this manuscript, Yegambaram et al., evaluate the overexpression of mitofusin 2 (Mfn2) in pulmonary arterial endothelial cells (PAECs) derived from a pulmonary hypertension model. Although the work seems interesting and the experiments well designed, the conclusions they reach are not supported by the established methods and the results presented. In this sense, the following points are requested to be clarified and explained in the manuscript:
How to distinguish overexpression of Mfn2 from restoring its expression in the evaluated model? since it is mentioned that these PAECs present reduced levels of Mfn2, however throughout the text the "overexpression" and "restoration" of Mfn2 levels are mentioned as if they were the same thing. The authors mention that "Three crucial findings arise from our studies": 1) "overexpression of Mfn2 in PAECs induces mitochondrial hyperfusion", 2) mitochondrial hyperfusion significantly attenuates mitochondrial bioenergetics and increases aerobic glycolysis". However, this mitochondrial hyperfusion is not demonstrated in any of their results. Although the presence of Mfn2 is highly associated with mitochondrial fusion, it also regulates other mechanisms, which are mentioned by the authors (line 54). If mitochondrial hyperfusion is not experimentally demonstrated in this manuscript, it is necessary to change the title and the sentences under discussion that assert this fact. 3) "this Warburg effect was induced by mt-ROS-mediated activation of HIF-1α protein levels". In this regard, no measurement is made to demonstrate that mt-ROS are mediating glycolysis or altering HIF-1α levels, remaining only speculation.
Response: We thank the reviewer for his important suggestions. We would like to point that this study aimed at understanding the consequence of mitofusin 2 (Mfn2) overexpression in control PAECs. Therefore, PAECs were isolated from control sheep. We used adenovirus-mediated transduction method to overexpress Mfn2 in PAECs and investigated if mitochondrial hyperfusion impacted cellular metabolism and mitochondrial bioenergetics. Regarding the reviewer’s inquiry, if Mfn2 mediates mitochondrial hyperfusion, we have added the new data (Figure 1E) in the revised manuscript. Results from our new fluorescent microscopy experiment (Figure 1E) show mitochondrial hyperfusion (increased mitochondrial aspect ratio) upon Mfn2 overexpression in PAECs. Further, we also demonstrate that Mfn2 overexpression increased apoptosis (Figure 1F) and decreased cell proliferation (Figure 1G).
minor comments:
chance mi-ROS by mt-ROS (line 225)
XF24 Analyzer and XFe24 analyzer are the same? (lines 341, 346)
line 367: What result were 3 groups evaluated on? that have been compared with ANOVA?
Response: We thank the reviewer for noting this. We have corrected the errors in the revised manuscript.
Reviewer 2 Report
Comments and Suggestions for Authors
Please see the attached file.

Comments on the Quality of English LanguagePlease see the attached file.
Author Response
Reviewer 2
Major comments:
- Title: The Authors should rewrite the title. It does not mention the role of Mfn2 in ECs
under physiological conditions, which the whole manuscript refers to. How strong is this
relation between Mfn2, mitochondrial dysfunction and PAH in view of the Mfn2 role in normal ECs? The Authors studied Mfn2 role in normal ECs but refer to PH which is in my opinion overinterpretation in view of data presented in this work. Furthermore, to claim the existence of a link between mito hyperfusion and metabolic reprogramming in PH, the Authors should include more studies on mitochondrial dynamics/morphology after targeting Mfn2 expression in ECs. Inclusion loss-of-function and gain-of-function studies in parallel might help to understand this complex/dynamic role of this mitochondrial enzyme. The Authors are encouraged to refer to the above mentioned comments/questions in the Discussion.
Response: We thank the reviewer for his important suggestions. Regarding the reviewer’s question about the relation between Mfn2, mitochondrial dysfunction and PAH, we want to emphasize that mitochondrial dynamics are balanced fission and fusion events that regulate mitochondrial morphology, and alteration in these events results in mitochondrial dysfunction. Mfn2 dysfunctions have been found to contribute to cardiovascular diseases including PH. Previously, other groups have reported that higher Mfn2 expression suppressed PH. However, in this study, we want to emphasize that higher Mfn2 expression in PAECs affects cellular metabolism and mitochondrial bioenergetics.
For the mitochondrial dynamics/morphology after targeting Mfn2 expression in ECs, we have added the new data (Figure 1E) in the revised manuscript that shows an increased mitochondrial aspect ratio upon Mfn2 overexpression in PAECs. The new experimental data, results, and discussion are included in the revised manuscript.
- Materials and Methods, the description of the WB and Seahorse protocols is not sufficient to repeat the experiments. Please see the comments below.
Response: We thank the reviewer for suggesting this. We have added detailed protocols in the materials and methods section.
- Lane 340-342: In the given reference [4], Glycolysis Stress Test is not described. Please include the protocol in the manuscript. What Seahorse medium was used? Please provide the composition (what was glucose concentration? Were other substrates added, like sodium pyruvate or L-glutamine. If yes, what was the final concentration)? Did the Authors compared the Mfn2 impact on mitochondrial bioenergetics/glycolysis in the absence vs. in the presence of L-glutamine? It is interesting as glutaminolysis is a driver of vascular remodeling in PAH. The Authors should also include when the cells were transduced prior Seahorse analysis, namely were they first seeded on Seahorse culture plate followed by adenoviral transduction? Or first transduced and afterwards re-splitted?
Response: We thank the reviewer for noting this. The revised manuscript includes detailed information about the reagents and step-by-step methods for the Seahorse XF assays.
Regarding the reviewer’s enquiry if Mfn2 impacted glutaminolysis in PAECs, we performed 13C-Glutamine flux analysis. We have added the new data (Figure 3) in the revised manuscript. Our 13C-Glutamine flux analysis revealed the absence of glutamine oxidation and reductive carboxylation changes in MFN2 over-expressing PAEC.
We have added a detailed protocol for adenoviral transduction in the materials and methods section.
- Lane 313: Please provide detailed information for the cell transfection protocol: DNA
amount per cm2, transfection reagent type and amount.
Response: We have included a detailed protocol for adenoviral transduction in the materials and methods section in the revised manuscript.
- Figure 4 and 5: Please provide a representative Seahorse graph from the Cell Mito Stress test to reflect the data presented on the graphs suggesting that also mitochondrial respiration parameters are changed after Mfn2 overexpression (Figure 5B-F). The representative Seahorse graph suggests there are no changes in basal respiration, ATP production, and a very slight change in maximal respiration and non-mitochondrial respiration. The Authors report that difference is statistically significant which may simply result from the data input as values are derived from n=10. The Author should more carefully interpret such small changes and consequently draw the final conclusions. Perhaps Mfn2 predominantly is involved in Warburg effects generation of a hyper-glycolytic phenotype of ECs without substantial impact on mitochondrial respiration. The oxidation of other substrates like glutamine or fatty acids may be affected to a larger extent. Please discuss.
Response: We thank the reviewer for suggesting this. We have provided the representative Seahorse graph from the Cell Mito Stress and Glycolysis tests in the revised manuscript. Since we observed increased apoptosis (Figure 1F) and reduced cell proliferation (Figure 1G) upon Mfn2 overexpression, we have normalized the seahorse data to protein concentration. We have added these new data to Figure 5 and 6 in the revised manuscript. The new normalized data revealed more significant differences between the Control and Ad-Mfn2 groups.
- The data in Figures 4 and 5 are expressed as mpH/min or pmol/min indicating that the
data were not normalized. The normalization (to protein or cell number) should be included and the statistical analysis performed again. The Seahorse data normalization is of a great importance because the cells (control vs. Mfn2 vector transfected) could be not seeded in the same number or/and could display a different rate of proliferation? Did the Authors analyze if Mfn2 overexpression affects ECs proliferation?
Response: We thank the reviewer for suggesting this. Indeed, we observed increased apoptosis (Figure 1F) and reduced cell proliferation (Figure 1G) upon Mfn2 overexpression. Therefore, we have normalized the seahorse data to protein concentration. We have added these new data to Figures 5 and 6 in the revised manuscript.
- Figure 1B: The given WB image does not indicate the Mfn1 increase after Mfn2 overexpression. Please revise.
Response: We thank the reviewer for noting this. We reanalyzed the Mfn1 band intensity in the Western blot. After reanalysis, we observed a slight increase in Mfn1 protein level upon Mfn2 overexpression. We have provided the original blot for verification (uploaded as Supporting Information files).
- Please provide the list of antibodies used in the study with the information enabling the
repetition of the experiments: catalog number, supplier, dilution (this could be a supplementary table).
Response: We have included detailed information about the reagents in the materials and method section in the revised manuscript.
- The ECs were isolated from lambs. The number of the animal ethics committee approval and appropriate statement should be included in the manuscript.
Response: We have included the Animal and ethics statement in the revised manuscript.
- Figure captions (a general comment): it is not necessary to repeat the results description/interpretation in the figure caption, e.g. Lane 107-111. Please revise and correct in the whole manuscript.
Response: We thank the reviewer for noting this. We have updated the figures captions in revised manuscript.
- Hyperglycolytic EC metabolism is tighly linked with resistance to apoptosis and increased proliferation of these cells. Did the Authors study apoptosis and proliferation (cell cycle) in normal PAECs overexpressing Mfn2? It would explain in detail the presented finding and perhaps the Mfn2 contradictory role in PAH and normal PAECs.
Response: We have included new figures in the revised manuscript. Indeed, we observed increased apoptosis (Figure 1F) and reduced cell proliferation (Figure 1G) upon Mfn2 overexpression. We want to emphasize that Mfn2 overexpression decreased cell proliferation in Control PAECs (observed in this study) and in PAH pulmonary arterial smooth muscle cells (reported previously by Ryan J. et al., PMID: 23449689).
- As defined in the Introduction, the mitochondrial dysfunction and hyperfusion is characterized by the alterations in mitochondrial activity and morphology. Was the mitochondrial morphology/mass changed upon Mfn2 overexpression? The Authors should include mitochondria morphology analysis, e.g., staining with monoclonal anti-mitochondria antibody, electron miscroscopy or labeling with MitoTracker.
Response: We have added the new data (Figure 1E) in the revised manuscript. Using fluorescent microscopy, we observed that Mfn2 overexpression increased the mitochondrial aspect ratio in PAECs, indicating increased mitochondrial fusion.
- What may be factors/mechanisms for contradictory roles of Mfn2 in normal and pathological (PH) ECs? Are there other mitochondrial enzymes identified to be involved in PH with a similar complex (twisted) nature?
Response: Although limited information is available linking dysregulation of fusion and fission mediators to PH, most papers have reported increased Drp1-mediated mitochondrial fission in PH. We have recently published a paper highlighting the importance of mitochondrial fission and a Warburg phenotype of increased cellular glycolysis involved in the pathogenesis of PH (Lu Q. el al., PMID: 36841483). Mitochondrial fission is increased in PAEC isolated from a sheep model of PH induced by pulmonary overcirculation (Shunt PAEC). The contradictory roles of Mfn2 in normal and pathological (PH) ECs have been previously reported. While overexpression of Mfn2 attenuates mitochondrial fission and reduces cellular glycolysis in PAECs isolated from PH lamb model, overexpression of Mfn2 in control PAECs as seen in this study promotes mitochondrial fusion and alters the fission/fusion balance and disrupts the glycolytic flux. Fusion is an underappreciated regulator of cell proliferation as the initial term for Mfn-2 was “hyperplasia suppressor gene” due to its anti-proliferative effect when over-expressed. Thus, the decrease in MFN2 in PH lambs could be involved in the endothelial cell hyper-proliferation associated with PH, and we observe the opposite When MFN2 levels are increased beyond normal levels. It’s the “goldilocks syndrome”.
We are not aware of mitochondrial enzymes with this complex nature. However, this janus concept has been identified in PH concerning endothelin-1 signaling. The ET B-type receptor is a vasodilator on ECs but a vasoconstrictor when found on the SMC in PH.
- Unfortunately, the original blots are not included in the submission. Please provide them.
Response: We have provided the original blot (uploaded as Supporting Information files).
Minor comments:
- Lane 20: please replace “are” with “is”.
- In Introduction, please clearly define the term “mitochondrial hyperfusion”. When increased fusion becomes hyperfusion?
- Lane 49 and 50: please eliminate the repetition “critical”.
- Lane 82: what “Mfn2 therapy”? Please be more specific.
- Please write the names of metabolites in the manuscript using lower case letter, e.g.,
Lanes 124-127.
- Lane 165, 178: please change “max” with “maximum”.
- Lane 192-193: please rewrite the sentence, e.g., “Together, these data link the increase in mt-ROS production with HIF-α-mediated induction of the Warburg phenotype of PAECs”.
- Lane 214-2-16: please rewrite the sentence, eliminate repeating “restore”. It would be more appropriate to say “Restoring Mfn2 expression”.
- Lane 217: please replace “wanted” with “aimed”
- Lane 225: Please replace “mi-ROS” with “mt-ROS”.
- Lane 218-219: The Authors should highlight that their present study refers to normal
ECs.
- Lane 256: please replace “overexpressed” with “overexpressing”.
Round 2
Reviewer 1 Report
Comments and Suggestions for Authors
Thank you for addressing the most significant comment on this work, improving it considerably, however, what about this comment?:
3) "this Warburg effect was induced by mt-ROS-mediated activation of HIF-1α protein levels". In this regard, no measurement is made to demonstrate that mt-ROS are mediating glycolysis or altering HIF-1α levels, remaining only speculation
Author Response
3) "this Warburg effect was induced by mt-ROS-mediated activation of HIF-1α protein levels". In this regard, no measurement is made to demonstrate that mt-ROS are mediating glycolysis or altering HIF-1α levels, remaining only speculation
Response: We thank the reviewer for asking this comment. We have updated the discussion section in the revised manuscript.
Previously, we have reported that endothelin -1 mediated increase in mt-ROS activated the HIF-1α signaling pathway and also enhanced the protein levels of HIF-1–dependent downstream target genes, enolase-2 and glucose-6-phosphate dehydrogenase, which are known to be involved in glycolysis (Sun, X. et al. Endothelin-1 induces a glycolytic switch in pulmonary arterial endothelial cells via the mitochondrial translocation of endothelial nitric oxide synthase. 2014. Am J Respir Cell Mol Biol 50, 1084-1095). Further, treatment with mitochondrial-targeted antioxidant (MitoQ) not only decreased endothelin-1–induced mt-ROS in PAECs, but also attenuated the ROS-mediated increase in HIF-1–dependent promoter activity, suggesting that mt-ROS activates the HIF-1 signaling pathway.
In this study, we identified increased levels of mt-ROS induced by increased Mfn2 in PAECs that correlated with increased HIF-1α levels, suggesting that mt-ROS activates the HIF-1 signaling pathway. Further, our recent work has shown that normalizing mitochondrial network dynamics decreases mt-ROS-mediated increases in HIF-1α and attenuates cellular glycolysis (Lu, Q., et al. Nitration-mediated activation of the small GTPase RhoA stimulates cellular glycolysis through enhanced mitochondrial fission. 2023. J Biol Chem 299, 103067).
Reviewer 2 Report
Comments and Suggestions for Authors
The Authors have substantially improved the manuscript, addressing the Reviewer’s remarks and suggestions. They have included additional experiments in order to decipher in detail the Mfn2 function in hyperfusion and metabolic reprogramming of PAECs.
There are a few minor remarks that the Authors should address:
1. In PAH, initial EC apoptosis is followed by the emergence of apoptosis-resistant proliferating EC (https://www.ncbi.nlm.nih.gov/pmc/articles/PMC2768704/, https://erj.ersjournals.com/content/40/6/1555)
Please include this finding in the Introduction and discuss your results in the Discussion.
2. Based on the provided Mito Stress Test Seahorse graph, the maximum respiratory capacity seems to be not changed statistically upon Mfn2 overexpression in PAECs? We can see on the graph that after FCCP injection, the maximal OCR is the same for control (the first measurement time point) and Ad Mfn2 (the third measurement time point). Maximum respiratory capacity refers to the OCR value (the highest one) after FCCP minus non-mitochondrial respiration. Please revise the calculations.
3. Lane 56, 163, 169-173: please write the whole protein names and metabolites without the first letter capitalized (e.g., mitofusin 2 instead of Mitofusin 2; glutamine instead of Glutamine, aconitase 2 instead of Aconitase 2, ACO2 instead Aco2, etc.). Please revise in the whole manuscript.
4. The Authors are encouraged to improve the manuscript title by highlighting the significance of Mfn2, as suggested in the first review round.
Author Response
There are a few minor remarks that the Authors should address:
- In PAH, initial EC apoptosis is followed by the emergence of apoptosis-resistant proliferating EC (https://www.ncbi.nlm.nih.gov/pmc/articles/PMC2768704/,
https://erj.ersjournals.com/content/40/6/1555)
Please include this finding in the Introduction and discuss your results in the Discussion.
Response: We thank the reviewer for providing this reference paper. We have included the reference paper and updated the introduction and discussion sections in the revised manuscript.
- Based on the provided Mito Stress Test Seahorse graph, the maximum respiratory capacity seems to be not changed statistically upon Mfn2 overexpression in PAECs? We can see on the graph that after FCCP injection, the maximal OCR is the same for control (the first measurement time point) and Ad Mfn2 (the third measurement time point). Maximum respiratory capacity refers to the OCR value (the highest one) after FCCP minus non-mitochondrial respiration. Please revise the calculations.
Response: First, we want to clarify that these data are taken directly from the Agilent Wave software. However, as the reviewer suggested, we reanalyzed the data using the graphpad prism software. Our analysis confirms that the maximum respiratory capacity statistically differs between the control and Ad-Mfn2 groups (p-value = 0.01).
- Lane 56, 163, 169-173: please write the whole protein names and metabolites without the first letter capitalized (e.g., mitofusin 2 instead of Mitofusin 2; glutamine instead of Glutamine, aconitase 2 instead of Aconitase 2, ACO2 instead Aco2, etc.). Please revise in the whole manuscript.
Response: We have corrected the errors in the revised manuscript.
- The Authors are encouraged to improve the manuscript title by highlighting the significance of Mfn2, as suggested in the first review round.
Response: We have updated the title in the revised manuscript.